

# Seeing the Wood for the Trees: Active human-
# environmental interactions in arid northwest China
**Hui Shen[1,2], Robert N. Spengler[3,4], Xinying Zhou[1,2], Alison Betts[5], Peter Weiming**
**Jia[5], Keliang Zhao[1,2], Xiaoqiang Li[1,2]**
[1] Key Laboratory of Vertebrate Evolution and Human Origins, Institute of Vertebrate Paleontology and
Paleoanthropology, Chinese Academy of Sciences, Beijing, 100044, China
[2] University of Chinese Academy of Sciences, Beijing, 100049, China
[3] Domestication and Anthropogenic Evolution Research Group, Max Planck Institute of
Geoanthropology, Jena, 07745, Germany
[4] Department of Archaeology, Max Planck Institute of Geoanthropology, Jena, 07745, Germany
[5] Department of Archaeology, University of Sydney, Sydney, NSW 2006, Australia
Corresponding author: Xiaoqiang Li
email: lixiaoqiang@ivpp.ac.cn





**Abstract:** Due largely to demographic growth, agricultural populations during the
Holocene became increasingly more impactful ecosystem engineers. Multidisciplinary
research has revealed a deep history of human-environmental dynamics; however,
these pre-modern anthropogenic ecosystem transformations and cultural adaptions are
still poorly understood. Here, we synthesis anthracological data to explore the
complex array of human-environmental interactions in the regions of the prehistoric
Silk Road. Our results suggest that these ancient humans were not passively impacted
by environmental change, but rather they culturally adapted to, and in turn altered,
arid ecosystems. Underpinned by the establishment of complex agricultural systems
on the western Loess Plateau, people may have started to manage chestnut trees,
likely through conservation of economically significant species, as early as 4600 BP.
Since ca. 3500 BP, with the appearance of high-yielding wheat/barley farming in
Xinjiang and the Hexi Corridor, people appear to have been cultivating *Prunus* and
*Morus* trees. We also argue that people were transporting the preferred coniferous
woods over long distance to meet the need for fuel and timber. After 2500 BP, people
in our study area were making conscious selections between wood types for craft
production, and were also clearly cultivating a wide range of long-generation
perennials, showing a remarkable traditional knowledge tied into the arid
environment. At the same time, the data suggest that there was significant
deforestation throughout the chronology of occupation, including a rapid decline of
slow-growing spruce forests and riparian woodlands across the northwest China. The
wood charcoal dataset is publicly available at https://doi.org/10.5281/zenodo.8158277
(Shen et al., 2023).
**Keywords:** Human-environmental interaction, human adaption, fruit management,
deforestation, northwest China



## 1 Introduction

The extent of prehistoric anthropogenic environmental change, especially relating to the ways early agricultural practices reshaped terrestrial ecosystems, has been the subject of ongoing debate (Ruddiman, 2003, 2008; Zong et al., 2007; Asouti and Kabukcu, 2014; Asouti et al., 2015; Dong et al., 2020a, 2022a). Over the past decade, scholars have adopted big data approaches to understanding long-term anthropogenic changes to the Earth's surface (Zalasiewicz et al., 2017; ArchaeoGLOBE Project, 2019; Renn, 2020; Cowie et al., 2022). While humans have undoubtedly been reshaping environments since before the Holocene, the magnitude of these impacts following the adoption of agricultural economies increased immensely. During this process, people shifted their subsistence system from hunting-gathering to cereal cultivation and animal husbandry, and increasingly gained the ability to alter and adapt their ecological surroundings (Bellwood, 2005; Zeder, 2008; Zohary et al., 2012). During the fifth millennium BP, agricultural populations across Europe and Asia first came into contact via diffusion of crops, contributing to food globalization in prehistory (Sherratt, 2006; Jones et al., 2011; Dong et al., 2017, 2022b; Boivin et al., 2016; Liu et al., 2019; Zhou et al., 2020). The intermingling of millets, adapted for arid and short-season grasslands in northern China, with cereals, adapted for rainy season growth in arid southwest Asia, eventually facilitated a greater intensification of farming systems (Spengler 2019; Miller et al. 2016).

Mounting evidence shows that the development of farming systems was accompanied by a series of ecological and social changes, including deforestation, wild species loss, and demographic expansion (Bellwood, 2005; Weisdorf, 2005; Atahan et al., 2008; Kaplan et al., 2009; Bocquet-Appel, 2011; Fuller et al., 2011a;



Asouti et al., 2015; Ruddiman, 2013). For instance, the dispersal and expansion of
agriculture largely altered the natural geographic distributions of anthropophilic plants
(crops and weeds) and directly influenced vegetation communities worldwide (Vigne
et al., 2012; Fuller et al., 2011b; Crowther et al., 2016; Boivin et al., 2017; Spengler et
al., 2021). Forest clearing, either to increase the surface area of arable land or to
acquire wood for construction or fuel, has caused large-scale deforestation and created
a more open landscape (Zong et al., 2007; Atahan et al., 2008; Kaplan et al., 2009;
Innes et al., 2013; Zheng et al., 2021). Meanwhile, human-mediated management of
local woodlands encouraged the growth of fruit- and nut-bearing trees, shifting land-
use strategies from an emphasis on short-term returns of annual cereals to long-term
investment with delayed return crops (Fall et al., 2002; Janick, 2005; Miller and
Gross, 2011; Miller, 2013; Asouti and Kabukcu, 2014; Asouti et al., 2015). Today,
essentially all ecosystems on the planet are anthropogenic constructs, recognized
through the increasingly prominent use of the term Anthropocene (Crutzen, 2002;
Ruddiman, 2003, 2013; Monastersky, 2015).

Northwest China, the focus region of this paper, is of particular interest, because

it is located at the core of the ancient trade routes that are colloquially referred to as
the Silk Road and farmers in the region were the first to experiment with agricultural
crops from both West and East Asia (Wang et al., 2017; Dong et al., 2017, 2018,
2022b; Zhou et al., 2020; Li, 2021). Archaeobotanical data have pinpointed the broad
region and time period when humans first started to cultivated millets in East Asia.
Specifically, evidence from the Dadiwan site has revealed that broomcorn millet
cultivation began as early as the eighth millennium BP (Liu et al., 2004; Li, 2018),
and the gradual diffusion of broomcorn millet reached famers in the mountains of
Central Asia by 4500 BP (Spengler et al. 2014; Yatoo et al. 2020). The remains of





barley and wheat found at the Tongtian Cave site, have been dated to around 5200
BP, representing the earliest known southwest Asian cereals found in East Asia (Zhou
et al., 2020). In addition to long-distance exchange of cereals, this area also fostered
the trans-continental dispersals of sheep, goat, bronze-smelting technology, mudbrick-
manufacturing techniques, and a variety of other cultural attributes (Mei and Shell,
1991; Dodson et al., 2009; Li et al., 2011; Yang et al., 2017; Dong et al., 2017; Chen
et al., 2018; Ren et al., 2022). Additionally, most of this region is characterized by a
hyper-arid desert and fragile oasis ecosystem, which are especially vulnerable to
human activity, making it a prime zone for studying the interaction between early
agricultural societies and the environment.
Archaeologists and geologists working in this region have mainly focused their
attention on the relationship between climate change and Neolithic cultural
development, as well as anthropogenic impacts on regional ecosystems. These
scholars have argued that enhanced precipitation during the Late Yangshao (5500-
5000 BP), Majiayao type (5300-4800 BP), and Qijia (4200-3800 BP) periods played
an important role in the expansion of these early farmers (An et al., 2004; 2005, 2006;
Hou et al. 2009; Liu et al., 2010; Dong et al., 2012, 2013, 2016, 2020a). A reduction
in the number of archaeological sites during the gap between early and middle
Majiayao (4800-4400 BP), and the decline of the Qijia culture are thought to be a
response to increasingly aridity (Dong et al., 2012, 2013). Concurrent with these
changes, people were actively engaged in reshaping the landscape. For instance, a
wood charcoal study from the Hexi Corridor has suggested that prehistoric wood
collection led to a rapid reduction in local woodlands and a decline in woody plant
diversity (Shen et al., 2018). In a different study, an increase in large-scale fire
frequency was proposed based on micro carbon records from Tian'e Lake, which was



further correlated with high Cu content, suggesting the consequence of large-scale
bronze smelting activities (Dong et al., 2020b). However, relatively less attention has
been paid to how agriculture influenced the cultural responses and adaption strategies
employed in these arid environments. Meanwhile, scientific records are
geographically uneven, with regions, such as the Hexi Corridor, attracting
considerable attention, while few studies have targeted the vast area of Xinjiang,
leading to an incomplete picture of prehistoric human-environmental interactions
along the ancient Silk Road.

In this study, we present a comprehensive synthesis of wood charcoal records

from northwest China. As the result of incomplete burning, wood charcoal fragments
from archaeological sites shed light on the practices of local woody plant use (Asouti
and Austin, 2005; Marguerie and Hunot, 2007; Théry-Parisot et al., 2010). Since the
first charcoal analyse, beginning in the 1940s (Salysbury and Jane, 1940), the
application of reflected light microscopy has allowed the rapid identification of
charcoal, making it widely used in: 1) the reconstruction of firewood collection
strategies (Li et al., 2016; Shen et al., 2018; Kabukcu, 2017; Mas et al., 2021); 2)
elucidating the impacts that wood cutting had on local forests (Li et al., 2011; Asouti
et al., 2015; Knapp et al., 2015; Shen et al., 2018); 3) identifying compositions of
woody communities (Wang et al., 2014; Asouti et al., 2015; Allué and Zaidner, 2022;
Mas et al., 2022); and 4) determining fruit and/or nut tree management (Miller, 2013;
Asouti and Kabukcu, 2014; Shen and Li, 2021). Here, we seek to identify patterns in
wood charcoal recovered from seven archaeological sites in Xinjiang, which we
contrast with more than 30 other published regional records. We aim to explore
multiple perspectives on the complexities of human-environmental interactions within
the agricultural background, including the influence of farming and wood cutting on



woody vegetation change, as well as the strategies applied in response to climatic
aridification.
**2 Study area**
*2.1 Regional setting*
Our study focuses on the provinces of Xinjiang and Gansu, because of the important
roles people in this region played in exchange along the ancient Silk Road. This
region is characterized by montane ecoclines, including those of the Tianshan, Altai,
Altun, and Qilian mountains (Figure 1). Due to glacial snowmelt, alluvial plains are
widely distributed across the low-land basins, and fine-grained nutrients and water
brought by the runoff nourish a network of oases, especially within the Hexi Corridor
and Tarim Basin (Zheng et al., 2015). Climatically, mean annual precipitation (MAP)
is geographically uneven, due to difference in prevailing air masses. For the West
Loess Plateau, which is under the control of the Asian monsoons, MAP usually
exceeds 400 mm (https://data.cma.cn/). Water vapour carried by the westerlies mainly
concentrates in the Ili or Irtysh valleys and Junggar Basin, and the MAP sometimes
can reach more than 500 mm (Xiao et al., 2006; Zheng et al., 2015). In the Tarim
Basin and the Hexi Corridor, the MAP is usually less than 200 mm
(https://data.cma.cn/). Temperatures are also spatially and seasonally unevenly
distributed; likewise, the mean annual temperature in the Kunlun, Tianshan, and Altai
mountains is below zero, while that of the Turpan Basin is around 14°C (Chen, 2010).


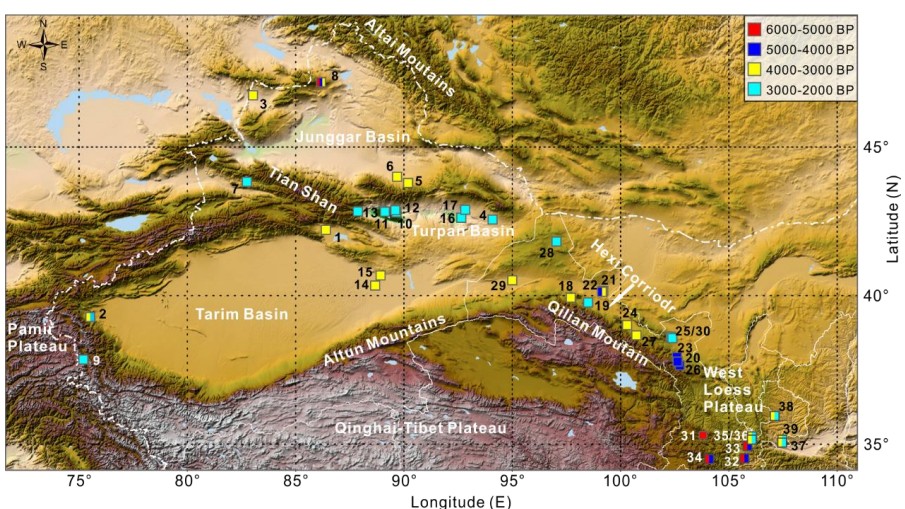

**Figure 1. The location of archaeological sites mentioned in this study**. 1 Xintala; 2 Wupaer; 3 Xiakalangguer; 4 Shirenzigou; 5 Sidaogou; 6 Xicaozi; 7 Qiongkeke; 8 Tongtian Cave; 9 Ji'rzankal; 10 Yanghai; 11 Jiayi; 12 Shengjindian; 13 Yuergou; 14 Xiaohe; 15 Gumugou; 16 South Aisikexiaer Cemetery; 17 Wupu; 18 Xihetan; 19 Zhaojiashuimo; 20 Huoshaogou 21 Huoshiliang; 22 Ganggangwa 23 Lifuzhai; 24 Xichengyi; 25 Sanjiao; 26 Mozuizi; 27 Donghuishan; 28 Jingbaoer; 29 Yingwoshu; 30 Sanjiaocheng; 31 Majiayao; 32 Xishanping; 33 Dadiwan; 34 Shannashuzha; 35 Daping; 36 Gaozhuang; 37 Jiangjiazui; 38 Laohuzui; 39 Qiaocun, the base map was obtained at https://www.ncei.noaa.gov/maps/grid-extract/.

Due to the arid climate, vegetation types here are characterized by expansive deserts (Xinjiang Integrated Expedition Team and Institute of Botany, 1978). Along the rivers in the low-land basins, riparian woodlands are mainly composed of *Populus*, *Elaeagnus*, *Ulmus*, and *Salix* (Chen, 2010). Within the montane belt, vegetation usually changes from grassland (dominated by *Stipa*), coniferous forest (mainly *Picea* and *Larix*), subalpine steppe (mainly *Stipa*), alpine meadows (including *Stipa*, *Carex*, and *Artemisia*), and alpine cushion vegetation (represented by *Androsace*, *Stellaria media*, and *Geranium wilfordii*), in banded ecoclines from lowest to highest elevation (Chen, 2010; Zheng et al., 2015; Xinjiang Integrated Expedition Team and Institute of Botany, 1978). Wild fruit and nut woodlands are distributed throughout the Tianshan Mountains, especially in the Ili valley, and the



main wild fruit trees include *Malus* sp., *Juglans regia*, and *Prunus* spp. (Chen, 2009;
Abudureheman et al., 2016).

## 2.2 Prehistoric cultures and agriculture

As an important cultural bridge connecting East and West Asia, northwest China has
fostered a variety of cultural communities. The early Neolithic cultures included the
Dadiwan and Yangshao, mainly distributed in southern Gansu (Institute of Cultural
Relics and Archaeology of Gansu, 2006). Later, people with material culture ascribed
to the Majiayao expanded quickly into the Hexi Corridor around 4800 BP (Xie, 2002;
Dong et al., 2020b). From 4000-3000 BP, the main archaeological cultures in Gansu
consisted of the Xichengyi, Qijia, Siba, and Dongjiatai (Li et al., 2010), and the
Shanma and Shajing cultures gradually developed after 3000 BP (Li, 2009; Gansu
Provincial Institute of Cultural Relics and Archaeology et al., 2015). In Xinjiang, the
prehistoric peoples before 4000 BP were represented by material culture categorized
as the Afanasievo and Chemurchek (Shao, 2018). From 4000-3500 BP, the
Andronovo Culture expanded into western Xinjiang, and the Tianshanbeilu and
Xiaohe cultures occupied the eastern Tianshan and Tarim Basin, respectively (Mei
and Shell, 1999; Ruan, 2014; Jia et al., 2017; Shao and Zhang, 2019; Xinjiang
Institute of Cultural Relics and Archaeology, 2004, 2014). Since 3500 BP, cultural
communities have continually diversified, with more localized groups forming, like
Subeixi Culture in the Turpan Basin (Chen, 2002).
Archaeobotanical evidence shows that millet cultivation was already practiced
by ca. 7800-7350 BP (Liu et al., 2004; Li, 2018). By at least 5500 years ago, people
were engaging in an intensive intermixed crop-livestock system by integrating pig
maintenance and millet cultivation (Yang et al., 2022). From 5000-4000 BP, both East



Asia millets diffused into the Hexi Corridor, while agricultural practices in Xinjiang
were restricted to limited microenvironmental pockets (Zhou et al., 2016; Dong et al.,
2017, 2018, 2020b; Li, 2021). Since 4000 BP, mixed agricultural systems composed
of both East and southwest Asian crops became more prominent; although, barley and
wheat had reached northwest China about a millennium prior (Flad et al., 2010; Zhao
et al., 2013; Yang et al., 2014; Zhang et al., 2017; Zhou et al., 2016, 2020; Jiang et al.,
2017a, 2017b; Tian et al., 2021). Stable carbon isotope data also suggest that the
consumption of both $C_3$ and $C_4$ plants was widely practiced after 4000 BP (Liu et al.,
2014; Zhang et al., 2015; An et al., 2017; Wang et al., 2016, 2017; Ma et al., 2016;
Qu et al., 2018). Around 3700-3300 BP, wheat and barley gradually replaced the
millets, becoming the dominant crops within the Hexi Corridor (Zhou et al., 2016).
From 3300-2200 BP, agriculture in Xinjiang gradually developed into something
more complex and spread to larger areas and more diverse ecozones, as evidenced by
the diversification of crops, and the appearance of irrigation technology and various
types of farming tools (Li, 2021). Meanwhile, secondary crops, such as *Vitis vinifera*
and *Ziziphus jujuba*, appeared more widely after ca. 2500 BP, indicating a strong
concept of land tenure associated with the development of agriculture (Jiang et al.,
2009, 2013; Li, 2021)
**3 Archaeobotanical Data and Chronology**
***3.1 Chronology of the archaeological sites***
In this study, we present data from seven archaeological sites and have developed a
chronology based on AMS [14]C dating through the Beta Analytic Testing Laboratory
and Australian Nuclear Science and Technology Organisation. For dating, we focused





on wheat seeds and wood charcoal, and the calibrated ages were generated using
Oxcal 4.4 with IntCal20 (Table 1 and Figure 2) (Reimer et al., 2020). The dating
results show that the seven archaeological sites cover a time span between 3900 and
2000 BP, and the oldest dates come from Xintala, at ca. 3900-3500 BP. The
Xiakalangguer, Sidagou, Xicaozi, and Qiongkeke sites fall in to the period of 3500-
3000 BP. The chronology for Shirenzigou covers roughly 2700-2000 BP. At Wupaer,
we collected wood charcoal samples from two sections, S1 and S3, and the date of the
S3 section is about 2900-2800 BP. The S1 section shows two different timespans,
specifically ca. 3400-3300 BP and 2500-2300 BP.
**Table 1. Dates for the seven archaeological sites in this study.**

| Site | Latitude | Longitude | Culture | Lab no. | Material | Date (BP) | Calibrated date (2δ, BP) | References |
|---|---|---|---|---|---|---|---|---|
| Xintala | 42.22 | 86.39 | Xintala type | OZM448 | charcoal | 3395±30 | 3815-3561 | Zhao et al., 2013 |
| | | | | OZM449 | charcoal | 3515±30 | 3877-3696 | |
| | | | | OZM450 | charcoal | 3335±30 | 3680-3469 | |
| | | | | OZM451 | wheat | 3460±35 | 3835-3593 | |
| | | | | OZL437 | wheat | 3515±50 | 3960-3642 | |
| Qiongkeke | 43.83 | 82.75 | Andronovo | Beta-642945 | charcoal | 3220±30 | 3482-3375 | this study |
| | | | | Beta-642946 | charcoal | 3320±30 | 3591-3458 | |
| Xiakalangguer | 46.74 | 83.03 | Andronovo | Beta-642943 | charcoal | 3140±30 | 3447-3327 | |
| | | | | Beta-642944 | charcoal | 3070±30 | 3365-3209 | |
| Sidaogou | 43.79 | 90.19 | Nanwan type | OZK664 | wheat | 3030±50 | 3362-3075 | Dodson et al., 2013 |
| | | | | OZK665 | wheat | 3080±60 | 3445-3080 | |
| Xicaozi | 44.00 | 89.68 | Unknown | OZM674 | wheat | 2975±45 | 3331-2997 | |
| Wupaer | 39.28 | 75.52 | Wupaer | Beta-642939 | charcoal | 3160±30 | 3451-3339 | this study |
| | | | | Beta-642940 | charcoal | 2450±30 | 2544-2361 | |
| | | | | Beta-642941 | charcoal | 2420±30 | 2515-2351 | |
| | | | | Beta-642942 | charcoal | 2800±30 | 2967-2844 | |
| Shirenzigou | 42.56 | 94.09 | Shirenzigou type | Beta-642947 | charcoal | 2350±30 | 2466-2329 | |
| | | | | Beta-642948 | charcoal | 2180±30 | 2313-2099 | |
| | | | | Beta-642949 | charcoal | 2150±30 | 2178-2041 | |
| | | | | Beta-642950 | charcoal | 2470±30 | 2715-2414 | |



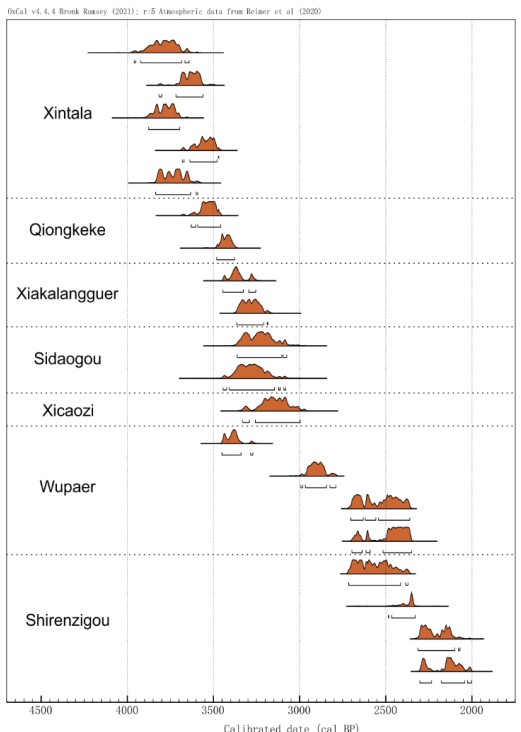


**Figure 2. The chronology of seven archaeologic sites in this study.**

### *3.2 Wood charcoal assemblages*

The identification of wood charcoal was accomplished via scanning electron
microscope, with 2,960 fragments of charcoal analysed and reported here (Appendix
A). Three of the sites are located in oases and wood charcoal assemblages show clear
similarities, with a dominance of *Tamarix* wood (Figure 3). In sediment from Xintala,
we identified 878 wood charcoal fragments, with *Tamarix* accounting for 74-95%.
*Elaeagnus angustifolia* increased across the chronology and reached its highest level
(13%) in the latest layer. There were limited occurrences of *Populus*, *Salix* and cf.
*Nitraria*. Wood charcoal from Wupaer also shows an abundance of *Tamarix* (ca.
80%), followed by fragments of *Populus*, *Salix*, and Chenopodioideae. Fruit tree
remains include *Prunus*, usually less than 3% in abundance. At the Xiakalangguer
site, *Salix* and *Tamarix* account for 44 and 28% of the assemblage respectively,
followed by Chenopodioideae (17%). A small number of fragments of *Betula* and
*Prunus* were also identified.

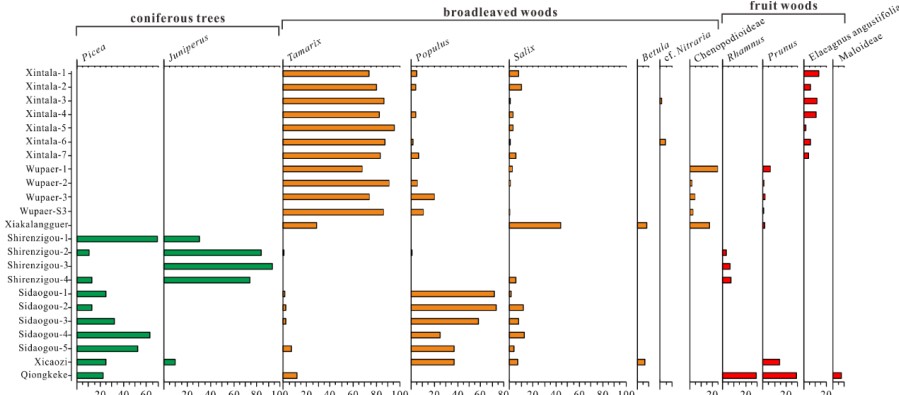


**Figure 3. Wood charcoal assemblages from seven archaeological sites in northwest China.**

In the eastern Tianshan, wood charcoal from three sites revealed an abundance of

coniferous wood fragments. At Shirenzigou, wood charcoal fragments from cultural
strata included *Picea*, *Juniperus*, *Tamarix*, *Populus*, *Salix*, and *Rhamnus*, with
conifers accounting for over 90% of the fragments. However, 14 wood samples taken
from coffins suggest that they are all made from coniferous woods, including *Picea*
(11) and *Juniperus* (3). At Sidaogou, wood charcoal from five samples was dominated
by *Picea* and *Populus*, followed by *Salix* and *Tamarix*. Progressively over time, *Picea*
fragments decreased from 52% to less than 20%, while *Populus* increased quickly
from 37% to over 70%. Similarly, *Picea* and *Populus* also constituted a dominant
percentage of the Xicaozi assemblage and the other taxa only cover a small
percentage, represented by *Prunus*, *Juniperus*, *Salix,* and *Betula*. The Qiongkeke site
is located in the Ili Valley, with five taxa identified among 229 wood charcoal



fragments. *Prunus* and *Rhamnus* account for 30% each. The proportion of *Picea* is
around 20%, followed by *Tamarix* and Maloideae.
In addition, we compiled wood charcoal data from published studies. In the Altai
Mountains, wood charcoal from Tongtian Cave indicates that people widely collected
*Larix*, *Picea*, *Betula*, *Populus*, *Salix*, Maloideae, and *Prunus* (Zhou et al., 2020). On
the Pamir Plateau, the data we have assembled from the Ji'rzankal Cemetery show
that *Populus* was used for making fire tools, *Betula* for wooden plates, *Salix* for
wooden sticks, *Juniperus* for fire altars, and *Lonicera* for arrow shafts (Shen et al.,
2015). Similarly, in the Turpan Basin, *Populus* was also selected for making fire tools
at the Yanghai Cemetery, and there was selective use of a variety of other woods,
including *Picea*, *Spiraea*, *Tamarix*, *Betula*, *Morus*, *Salix*, *Clematis,* and *Vitis vinifera*
(Jiang, 2022). *Lonicera* was also used for arrow shafts and composite bows at the
Jiayi and Shengjindian cemeteries (Nong et al., 2023). *Picea* was widely used at
Yuergou for coffin manufacture and firewood (Jiang et al., 2013). While in the Tarim
and Hami basins, *Populus* and *Tamarix* were largely used for coffins and wooden
utensils, as revealed by studies at the Xiaohe, Gumugou, South Aisikexiaer, and
Wupu cemeteries (Institute of Cultural Relics and Archaeology of Xinjiang, 2007,
Zhang et al., 2017, 2019; Wang et al., 2021).
In the Hexi Corridor, *Picea* and/or *Juniperus* constituted the dominant portion of
wood charcoal fragments in sites located near the Qilian Mountains, such as Xihetan
and Zhaojiashuimo (Shen et al., 2018). While wood charcoal from oasis sites, like
Huoshaogou, Huoshiliang, and Ganggangwa, also record the abundance of *Tamarix*,
and woody Polygonaceae and *Salix* disappear from later phases of Huoshiliang,
presumably due to over harvesting for fuel (Shen et al., 2018, Li et al., 2011). The





other sites in this area are characterized by abundant broadleaved taxa, with a small
percentage of coniferous wood fragments, such as at the Lifuzhai, Xichengyi, and
Sanjiao sites (Wang et al., 2014; Shen et al., 2018; Liu et al., 2019). Meanwhile, wood
charcoal assemblages from the Mozuizi and Donghuishan sites suggest a rapid decline
of local wood sources, including those of *Picea*, Maloideae*,* and *Betula* (Shen et al.,
2018). Additionally, an abundance of *Prunus* wood fragments was found in these two
sites, and people might have transported *Picea* wood over long distances to burn at
Donghuishan (Shen et al., 2018). The long-distance transport of *Picea* and *Pinus* was
also recognized in the assemblage from the Jingbaoer jade mine (Liu et al., 2021). At
the Yingwoshu and Sanjiaocheng sites, abundant *Morus* wood fragments were
identified, possibly indicating the early cultivation of mulberry (Shen et al., 2018).

As with the Hexi Corridor, wood taxa recovered from the western Loess Plateau

also suggest a quick decline in the abundance of *Picea*, notably from 37% to less than
4% at Majiayao (Shen et al., 2021). In the assemblage from Xishanping, *Picea*,
*Betula*, *Acer*, and *Quercus* decreased markedly after 4600 BP, and *Picea* declined
from a peak value of 28% to less than 5%, while Bambusoideae increased sharply (Li
et al., 2012). The sudden spike on abundance of bamboo is thought to be due to rapid
successional colonization after significant deforestation or clearing of woody
competitive species. Meanwhile, fruit trees, including *Castanea*, *Prunus* (what the
wood specialists in this study called *Cerasus* and *Padus),* and *Diospyros* expressed a
considerable increase in abundance (Li et al., 2012). The use of fruit tree wood was
also recognized in the Dadiwan, Shannashuzha, Daping, and Gaozhuang sites, with
the abundance of *Prunus* (these researchers subdivided this group into *Prunus* and
*Padus*, which we have clumped together in this study for consistency), Maloideae,
and *Ziziphus* (Sun et al., 2013; An et al., 2014; Li et al., 2017).



## 4 Discussions and Conclusion

### *4.1 Wood collection strategies and the transport of conifers*

As the result of wood burning, wood charcoal provides insights into the decision-making process regarding the collection of fuel. In this study, we found that wood charcoal assemblages from all oasis sites were dominated by *Tamarix*. Most species from the *Tamarix* genus are deciduous shrubs, generally 2-5 meters high, with slender and soft branches (Yang and Gaskin, 2012). The twigs are often browsed by sheep, camel, and donkey, and the branches can serve as a rapidly-regenerating fuel (Editorial Board of Flora of China, CAS, 1990). Therefore, this widely-distributed, arid-tolerant, and rapid-growing shrubby *Tamarix*, might constitute the best fuel for ancient oases groups. For the archaeological sites located in mountainous areas, wood fragments from coniferous trees are more prevalent. For example, abundant *Picea* and *Juniperus* wood fragments were found at Shirenzigou in the eastern Tianshan. Similarly, *Picea*/*Juniperus* constitutes the dominant portion of the fragments from sites near the Qilian Mountains (Shen et al., 2018). All of the assemblages show that people were largely opportunistic in their choices and the availability of wood sources played a key role in the wood collection strategies.

Additionally, as wood resources in arid northwest China are relatively limited, coping with localized wood shortages would have been an issue that people inevitably dealt with. Among these wood charcoal assemblages, we found that there are some fragments of coniferous woods that likely represent people traveling over long distances on collection trips. The earliest known evidence might come from Donghuishan (3700-3400 BP), in which *Picea* charcoal experienced a sharp decrease



and then suddenly increased to its highest level (Shen et al., 2018). Given that spruce
forests are very slow to regenerate, the sudden increase of spruce fragments was likely
the result of long-distance collection from the Qilian Mountains (Shen et al., 2018).
Generally, spruce wood has preferential properties, as its timber is straight and tall,
and easily worked, presumably contributing to the selection and transportation of this
specific species. Since 2500 BP, the long-distance collection of coniferous woods
seems to have been a more regular activity, as evidenced at the Jingbaoer jade mine,
where *Picea* and *Pinus* wood fragments are recovered well outside their natural
ecological distribution (Liu et al., 2021). In the Turpan Basin, *Picea* wood fragments
were found in sediments from a series of Subeixi sites, which may have been
collected from the Tianshan Mountains (Jiang et al., 2013; Jiang, 2022).
In addition to noting the likely long-distance collection of coniferous woods, the
abundance of conifers in most of our study sites hints to the likelihood that people
might also have a preference for this specific wood type. At Sidaogou, spruce wood
fragments comprise more than 60% of the total fragment assemblage. Similarly,
charcoal from Majiayao recorded spruce fragments as the most used taxon right from
the onset of when people settled down at the location (Shen et al., 2021). Meanwhile,
the exclusive use of coniferous wood for coffin construction is also recognizable in
this study. At Shirenzigou, the analysis of 14 wooden coffins show that they were all
made of coniferous woods. However, in sediments from the site, we found a variety
of carbonized wood types, including *Tamarix*, *Populus*, *Rhamnus*, *Salix*, etc.
Historically, a preference towards coniferous woods is widely noted in ancient China
(Ding, 2022), and archaeological wood studies in Central Asia have also noted similar
patterns (Spengler and Willcox 2013). Many ethnographic and historical references to
ritual juniper twig burning as incense are noted from across Inner Asia. The fact that



the wooden coffins at Shirenzigou are all constructed from conifers, suggests that the
ritual significance of the resinous trees may stretch much further back in time.
Ultimately, we conclude that an awareness of the properties and special meaning of
these woods probably plays a key role in their wide use.

### 4.2 Collection and cultivation of fruit trees

In addition to the prehistoric expansion of agricultural systems, the significant
amounts of fruit wood fragments in our study may imply that the anthropogenic
processes were increasing the density of fruit trees near human settlements. Presently,
scholars continue to grapple with the question of what evidence is necessary to
differentiate between wild foraging, conservation of economically significant trees
and low-investment cultivation of wild populations (Dal Martello et al., 2023). In our
study, fruit wood fragments before 4600 BP were usually found in low percentages,
indicating limited collection of seasonally available wild fruits (Sun et al., 2013; Li et
al., 2017; Shen et al., 2021). Roughly between 4600-4300 BP, *Castanea*, *Prunus,* and
*Diospyros* charcoal shows a rapid increase in abundance at Xishanping on the western
Loess Plateau (Li et al., 2012). Pollen data at this time also demonstrates that
*Castanea* became the dominant broadleaved taxon, which is quite different from the
reconstructed natural vegetation, likely indicating the management of wild chestnut
forests or at least that humans were choosing not to cut these trees down, increasing
their populations (Li et al., 2007). Also, archaeobotanical records at this site illustrate
that a complex agricultural system based on a variety of crops, including millets, rice,
oats, soybean, and buckwheat, appeared synchronously with the management of
chestnut. This cooccurrence probably suggests that the exploitation of secondary
crops was closely related to and underpinned by the well-organized agricultural



system.
During the period from 4300 to 3500 years ago, there is an increase in the
abundance of fruit wood remains in Xinjiang and the Hexi Corridor. For example,
*Elaeagnus angustifolia* charcoal was found throughout the whole section and shows a
gradually increasing trend at Xintala. In the Hexi Corridor, *Prunus* wood fragments
were found in great abundance at Mozuizi and Donghuishan, far higher than its
percentage is believed to have been in the natural vegetation, possibly showing an
intensive collection of *Prunus* (Shen et al., 2019). However, there is no clear sign of
fruit management during this period, given that a wide range of wild fruit types, such
as *Nitraria* and *Cotoneaster* were also widely exploited (Zhou et al., 2016; Shen et al.,
2019). Meanwhile, previous studies show that, although a mixed agricultural system
consisting of both millets, wheat, and barley existed in Xinjiang and the Hexi
Corridor after 4000 BP, people still relied heavily on animal herding and/or feeding
(Dong et al., 2020b; Li, 2021).
From 3500-2500 BP, the cultivation or maintenance of *Prunus* and *Morus* trees
was probably adopted into the agricultural system. As in Wupaer, located in the
Kashgar oasis, the presence of *Prunus* charcoal remains is beyond its natural
distribution and the climatic conditions around the site are not suitable for the growth
of *Prunus*, likely resulted from anthropogenic planting. On the other hand,
considering that the distribution of wild *Prunus* trees had largely shrunk or even
disappeared presumably due to long-term human activity, we should still be cautious
about this conclusion. Almost at the same time, people in the Hexi Corridor probably
also started engaging in horticultural practices, supported by the abundant discovery
of *Morus* charcoal (Shen et al., 2019). Synchronously, a high-yield wheat and barley



farming system was developed in the Hexi Corridor (Zhou et al., 2012), and a more
intensified agricultural system developed in Xinjiang (Li, 2021), likely providing a
fundamental basis for the exploration of delayed-return perennial crops.
After 2500 BP, the cultivation of fruit trees was probably a widely practice in
northwest China. For instance, evidence from the Turpan Basin shows the presence of
*Morus* woods and *Vitis vinifera* stems at the Yanghai cemetery (Jiang, 2022; Jiang et
al., 2009), *Vitis vinifera* seeds in the Shengjindian cemetery (Jiang et al., 2015), and
*Ziziphus jujuba* stones in the Yuergou site (Jiang et al., 2013). At the Sampula
cemetery, fruit, nut and seed types were more abundant, including *P. persica*, *P.*
*armeniaca*, *Juglans regia*, *Coix lacryma-jobi*, etc. (Jiang et al., 2008). The appearance
of such a rich and diverse array of fruit crops indicates that people in northwest China
had developed a complex indigenous knowledge to survive in this hyper arid
environment and conducted more and more frequent exchange across the Eurasian
continent.

### *4.3 Indigenous knowledge of plant resources*

Due to the extreme arid climate, wooden objects found in our study area are usually
well-preserved and the data suggest that people might have also captured the
knowledge of deliberately selecting certain types of woods when making various
utensils. For example, within the Subeixi groups in the Turpan Basin, *Lonicera* was
harvested from wild stands for making arrow shafts at Jiayi and Shengjingdian (Nong
et al., 2023). At the Yanghai cemetery, *Betula* was selected for making dippers or
ladles, for its rigidity; flammable *Populus* and *Picea* were used for fire tool
manufacture (Jiang et al., 2018, 2021). People at this time also used *Lithospermum*
*officinale* seeds for decoration (Jiang et al., 2007a), *Nitraria tangutorum* for making



necklace (Jiang, 2022), and *Cannabis* for ritualized consumption and/or medical
purposes, as revealed in both the Turpan Basin (Jiang et al., 2006, 2007b, 2016) and
the Pamir Plateau (Ren et al., 2019).
Similarly, on the Pamir Plateau, *Betula*, which has high rigidity and density, and
homogeneous texture, was selected for making wooden plates (Shen et al., 2015).
Additionally, it appears that people specifically chose flammable *Populus* wood to
make fire tools; *Salix*, with long and straight branches, was used for fashioning
wooden sticks; sweet-scented *Juniperus* was the preferred choice for making fire
altars, and *Lonicera* was selected for arrow shaft manufacture. Such conscious
utilization of different wood properties illustrates the ingenuity of these ancient
people. Although the current archaeobotanical research related to wooden utensils is
still limited, studies from the Turpan Basin and the Pamir Plateau clearly suggest that
the conscious selection of wood types for specific properties was a particularly
pronounced practice after 2500 BP, especially among cultural contexts of a well-
established agriculture base with millets, wheat, and barley. Meanwhile, the
appearance of horticulture based on a variety of secondary crops at the time indicated
a more settled lifestyle, which might provide opportunities for prehistoric people to
fully explore and make the best use of the indigenous plant resources.

### 460 *4.4 Anthropogenic deforestation*

Presumably via slash and burn agriculture, people have largely altered terrestrial
ecosystems across the globe (Zong et al., 2007; Schlütz et al., 2009; Li et al., 2009;
Neumann et al., 2012; Innes et al., 2013; Ma et al., 2020; Zheng et al., 2021). For
northwest China, wood charcoal data in this study show that, apart from diversified
cultural adaption, human-induced landscape alteration also occurred widely, not only




throughout the whole history of agricultural activity, but also across different
vegetation contexts. For example, wood charcoal data from Sidaogou in the eastern
Tianshan recorded a significant decrease in abundance of spruce wood fragments
(Figure 4). Meanwhile, *Tamarix* and *Salix* nearly disappeared in the later stage,
showing that wood cutting caused a sharp attenuation of spruce forests and
broadleaved woodland. Similarly, *Tamarix* charcoal from the Xintala section in the
Yanqi Oasis firstly increased and then decreased to its lowest level in the upper layer,
suggesting that continuous wood cutting resulted in the decline of *Tamarix* shrubs. At
the same time, *Populus* and *Salix* charcoal disappeared in the middle layer, implying
that local riparian woodlands were largely deforested.

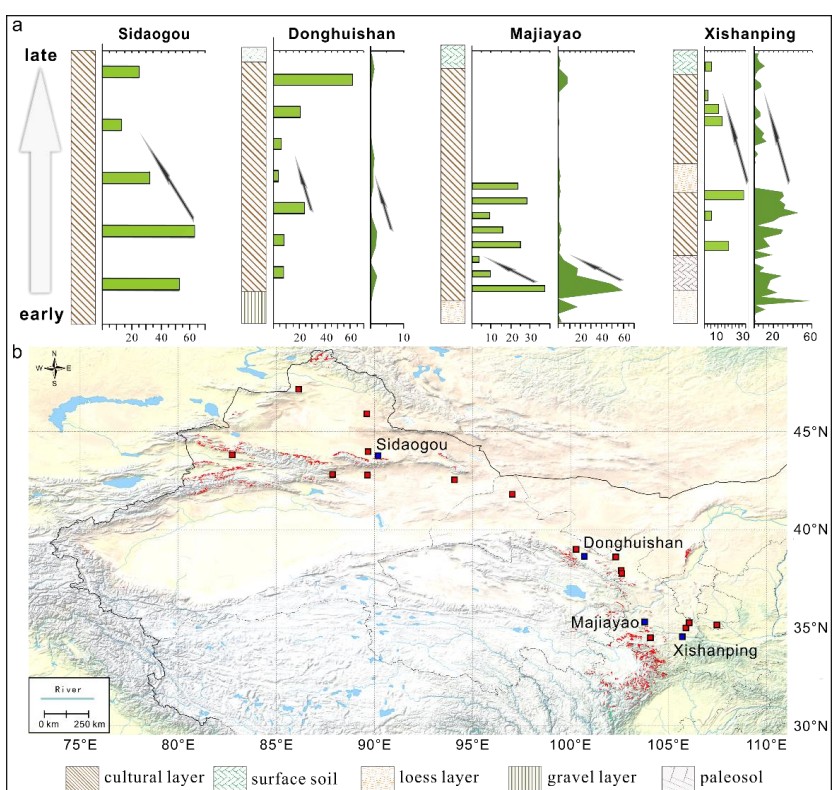


**Figure 4. The wood charcoal and pollen records show synchronous deforestation of spruce**



**forests across all of northwest China. (a) the change of *Picea* wood charcoal (bar) and pollen**
**(curve) from Sidaogou, Donghuishan (Zhou et al., 2012; Shen et al., 2018), Majiayao (Zhou,**
**2009; Shen et al., 2021), and Xishanping (Li et al., 2007, 2012). (b) the comparison of spruce**
**forests between prehistoric times and now, the squares represent archaeological sites with**
***Picea* charcoal remains and the red areas show the current distribution of spruce forests in**
**northwest China (after Hou, 2019).**
The Neolithic deforestation and reduction in range of spruce forests have also
been widely recognized across the western Loess Plateau and the Hexi Corridor. At
the Majiayao site, wood charcoal recorded the rapid decline of *Picea* during the early
stages of the site's occupation (Figure 4) (Shen et al., 2021). Not far from Majiayao,
wood charcoal from the Xishanping section revealed a similar pattern, with *Picea*,
*Betula*, *Acer*, *Ulmus*, and *Quercus*, illustrating a marked decrease after 4600 BP,
while Bambusoideae quickly colonized after the clearing of the original forest (Li et
al., 2012). In the Hexi Corridor, wood charcoal assemblages from the Mozuizi and
Donghuishan sites show a quick decline in plant diversity concurrent with human
settlement, and the percentage of *Picea* from Donghuishan recorded a sharp decrease
(Figure 4) (Shen et al, 2018). Similarly, wood charcoal fragments from Huoshiliang
show that *Salix* and Polygonaceae almost disappear, likely due to the large demand
for fuel used in bronze smelting activities (Li et al., 2011). Collectively, we interpret
the broader trend throughout all of these wood charcoal assemblages as revealing a
rather rapid process of deforestation across northwest China, especially shown in the
large-scale reduction in spruce forests. Our results are also supported by evidence
from pollen records, especially *Picea* pollen from Majiayao (Zhou, 2009), Xishanping
(Li et al., 2007), Donghuishan (Zhou et al., 2012), and other sections from the Loess
Plateau (Zhou and Li, 2011). All of these records document considerable reduction in





spruce forests (Figure 4). Today, the distribution of spruce forests has shrunk down to
a few constrained small forest patches (Figure 4).

**5 Data availability**

The datasets of archaeobotanical wood charcoal records in northwest China including
taxa types, absolute counts of wood charcoal fragments, and the locations and AMS
[14]C dates of each archaeological site are available at the open-access repository
Zenodo (Shen et al., 2023; https://doi.org/10.5281/zenodo.8158277).

**6 Summary**

The synthesis of wood charcoal data from nearly 40 archaeological sites shows that
prehistoric human-environmental interactions in northwest China were closely related
to the development of agriculture and considerably more complicated than previously
thought (Figure 5). Although anthropogenic deforestation occurred throughout the
whole period, most evidently relating to the decline of spruce forests, people also
actively applied a range of adaptive strategies to survive in this harsh environment. As
early as 4600 BP, people on the western Loess Plateau might have started managing
or at least conserving chestnut trees, likely underpinned by the development of a
complex agricultural system. Since ca. 3500 BP, with the appearance of high-yielding
agriculture based on wheat and barley in Xinjiang and the Hexi Corridor, people
appear to have been planting perennial tree crops, such as *Prunus* and *Morus*.
Additionally, they likely engaged in long-distance transportation of preferred woods,
specifically coniferous trees. After 2500 BP, people successfully mastered a wide
range of adaption strategies along the ancient Silk Road, as they began manufacturing
wooden utensils with conscious selection of wood properties. Moreover, the




consumption of a further diversity of fruit types, including grapes, signalled more
intensive horticultural practices and complex social structure.

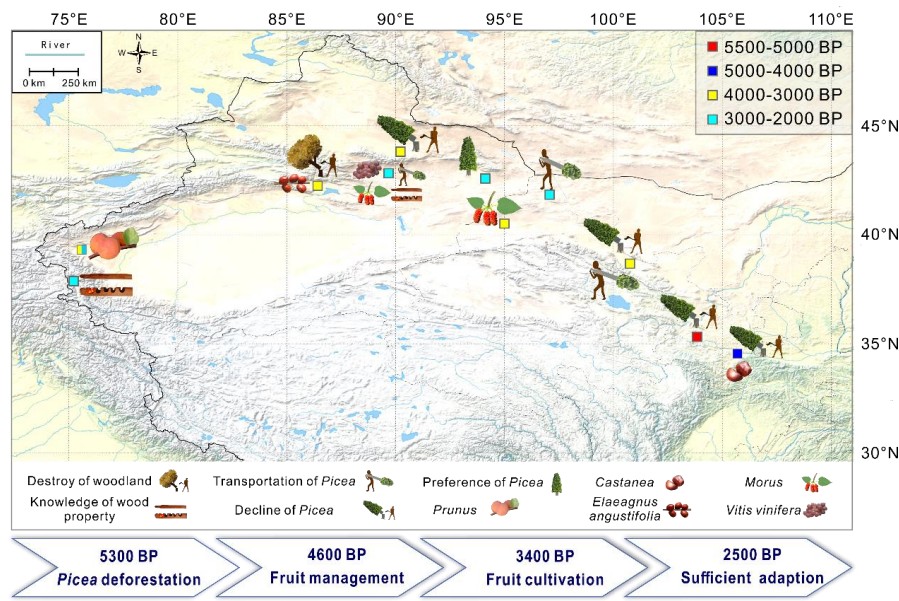


**Figure 5. A summary of prehistory human-environmental interactions in northwest China.**











**Appendix A**. The selected scanning electron microscopic images of wood charcoal in

Xinjiang. (a-c) *Picea*. (d-f) *Prunus*. (g-I) *Populus*. (j-l) *Tamarix*.

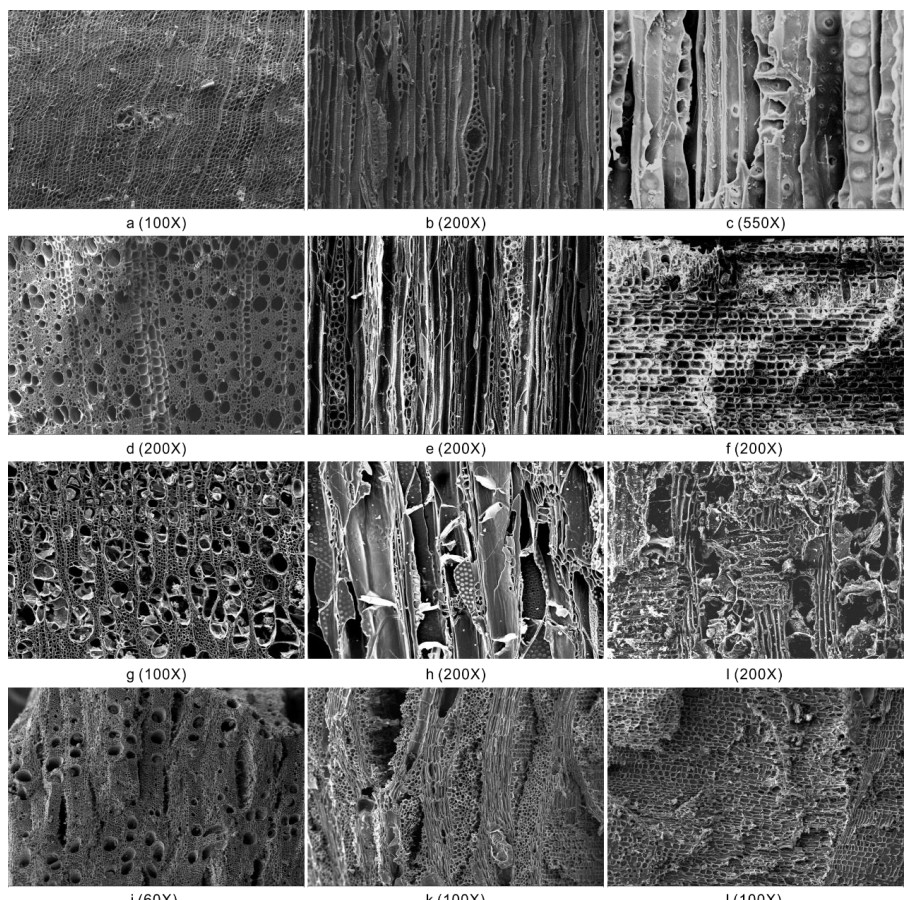

**Author contributions.** HS and XL designed the archaeobotanical dataset; HS was

responsible for construction of the database; HS performed numerical analyses and

organized the manuscript, and XZ, RS, PJ and AB revised the draft of the paper. All

authors discussed the results and contributed to the final paper.

**Competing interests.** The contact author has declared that none of the authors has

any competing interests.



**Acknowledgements.** We sincerely thank Ming Ji and Hongbin Zhang for their help in
the wood charcoal sample collection, and Nan Sun for her assistance with data
collection.

**Financial support.** This research has been supported by the National Natural Science
Foundation of China (grant no. 42002202), the Youth Innovation Promotion
Association of Chinese Academy of Sciences (grant no. 2022071), and the National
Key Research and Development Program of China (grant no. 2022YFF0801502).

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
