# Peer review of "Seeing the Wood for the Trees: Active human- # environmental interactions in arid northwest China"

_Earth System Science Data, 2023_

## Referee Comment (RC1)

Reviewer's comment

This manuscript reports human-environment interaction in northwestern China during the late Neolithic and Bronze Age. The study highlighted the impact of agriculture on human wood utilization in this area. On the whole, I believe that this manuscript has significant merit to warrant publication, in particular the amount of work done to reveal the history of human wood usage in northwestern China. However, there are still some issues with this manuscript, and should be revised. The major concerns are as follow:

Introduction:

I am uncertain what the exact question or hypothesis is of the authors. It seems to me like there was no clear expectation set forth at the beginning of the manuscript and that the authors just did a large-scale analysis with no clear plan. The question in this study is "Anthropogenic impact on environment" or "wood usage and its influencing factors" ? From line 120-122, it seems to discuss "wood usage and its influencing factors" ?

Discussion:

The author highlighted that anthropogenic impact on deforestation. However, has climate change also played the role? Because, the climate in northwestern China has become cold and dry since the late Neolithic. So if the deforestation was entirely caused by human wood collection, it should be ruled out the impact of climate change on woody vegetation, especially that of Spcure.

The pollen records in Fig 4 are from the archaeological sites? The captions of this figure should be more explained. If so, the similar changes in the records of charcoal and pollen do not indicate the woody degradation in northwestern China. Because pollen records from the section of archaeological site likely also show the trend of wood usage.

---

## Author Comment (AC1)

Reviewer's comment

This manuscript reports human-environment interaction in northwestern China during the late Neolithic and Bronze Age. The study highlighted the impact of agriculture on human wood utilization in this area. On the whole, I believe that this manuscript has significant merit to warrant publication, in particular the amount of work done to reveal the history of human wood usage in northwestern China. However, there are still some issues with this manuscript, and should be revised. The major concerns are as follow:

Introduction:

I am uncertain what the exact question or hypothesis is of the authors. It seems to me like there was no clear expectation set forth at the beginning of the manuscript and that the authors just did a large-scale analysis with no clear plan. The question in this study is "Anthropogenic impact on environment" or "wood usage and its influencing factors" ? From line 120-122, it seems to discuss "wood usage and its influencing factors" ?

We will provide a clearer set of goals or hypotheses relating to this paper in the introduction. Specifically, we sought to test the assumption that humans in the deeper past were changing forest communities; we also wanted to better understand how human culture allowed for better adaptations to a dynamic ecology over time along the ancient Silk Road. In many other parts of the world, it has been shown that when humans began to cultivate crops and engage in agriculture, the magnitude and complexity of their environmental interactions increased immensely. Northwestern China is located in the core area of the ancient Silk Road, and people in this area were the first to experiment with agricultural systems that integrated crops from both West and East Asia, making it an ideal zone for studying the interaction between early farmers and their environments. Over the past decade, scholars working in this region have mainly focused their attention on the relationship between climate change and Neolithic cultural development. To a lesser extent, scholars have studied anthropogenic impacts on regional ecosystems. Meanwhile, relatively less attention has been paid to the cultural responses and adaption strategies employed by early farmers in these arid environments.

Discussion:

The author highlighted that anthropogenic impact on deforestation. However, has climate change also played the role? Because, the climate in northwestern China has become cold and dry since the late Neolithic. So if the deforestation was entirely caused by human wood collection, it should be ruled out the impact of climate change on woody vegetation, especially that of Spruce.

The reviewer is, of course, correct that we should rule out the impact of climate change on woody vegetation here. In fact, this one of the core dilemmas that paleoenvironmental work grapples with; differentiating anthropogenic and natural environmental changes in the past is not easy, and indeed, not always possible. That said, in this study, in order to figure out the temporal change of vegetation and its

influencing factor(s), we focused on wood charcoal data from archaeological sites with continuous strata. Although we cannot totally differentiate between these factors, we can try to make correlations between paleoclimatic data and the archaeological record, for example:

Along the Tianshan mountains, wood charcoal data from Sidaogou (3400-3000 BP) recorded a significant decrease of spruce wood fragments, and *Tamarix* and *Salix* nearly disappeared in the later stage. Similarly, *Tamarix* charcoal from the Xintala (3900-3500 BP) section decreased to its lowest level in the upper layer, *Populus* and *Salix* charcoal disappeared in the middle layer. However, pollen records from Bosten and Balikun lakes showed a relatively stable climate during 3900-3500 BP, and a long-term increase of humidity after 3800 BP (Chen et al., 2006; Huang et al., 2009; An et al., 2012). Thus, we infer that the quick decrease in woody plants was resulted from anthropogenic exploration rather than climate change. Also, on the western Loess Plateau, the sudden disappearance of conifer trees after 4600 BP coincided with the appearance of wheat, barley and buckwheat in the Xishanping site, indicating intensive agriculture activities (Li et al., 2007). In the Majiayao section, the wood charcoal of *Picea* suddenly decreased from its highest level of nearly 40% to the lowest of less than 4% at around 5300-5100 BP. While, according to an analysis based on a high-resolution (~5 years) stalagmite sequence from the western Loess Plateau, there was no abrupt climate events during this time (Tan et al., 2020). In the Hexi Corridor, previous studies based on wood charcoal records also pointed to an anthropogenic impact rather than climatic change that caused a quick decline of riparian woodland and coniferous forest (Li et al., 2011; Shen et al., 2018). Collectively, we concluded that our wood charcoal revealed widespread deforestation starting with Neolithic agriculture.

References:

Chen, F. H., Huang, X. Z., Yang, M. L., Yang, X. L., Fan, Y. X., and Zhao, H.: Westerly dominated Holocene climate model in arid Central Asia—case study on Bosten Lake, Xinjiang, China, Quat. Sci., 26(6), 881-887, 2006 (in Chinese with English abstract).

Huang, X. Z., Chen, F. H., Fan, Y. X., and Yang, M. L.: Dry late-glacial and early Holocene climate in arid Central Asia indicated by lithological and palynological evidence from Bosten Lake, China, Quat. Int., 194, 19-27, https://doi.org/10.1016/j.quaint.2007.10.002, 2009.

An, C. B., Lu, Y., Zhao, J., Tao, S., Dong, W., Li, H., Jin, M., and Wang, Z.: A high-resolution record of Holocene environmental and climatic changes from Lake Balikun (Xinjiang, China): Implications for central Asia, Holocene, 22(1), 43-52, https://doi.org/10.1177/0959683611405244, 2012.

Li, X., Zhou, X., Zhou, J., Dodson, J., Zhang H., and Shang X.: The earliest archaeobiological evidence of the broadening agriculture in China recorded at Xishanping site in Gansu Province, Sci. China Ser. D-Earth Sci. 50, 1707-–1714, https://doi.org/10.1007/s11430-007-0066-0, 2007.

Tan, L., Li, Y., Wang, X., Cai, Y., Lin, F., Cheng, H., Ma, L., Sinha, A., and Edwards L.: Holocene monsoon change and abrupt events on the western Chinese Loess Plateau as revealed by accurately dated stalagmites, Geophys. Res. Lett., 46, e2020GL090273, https://doi.org/10.1029/2020GL090273, 2020.

Li, X., Sun, N., Dodson, J., Ji, M., Zhao, K., and Zhou, X.: The impact of early smelting on the environment of Huoshiliang in Hexi Corridor, NW China, as

recorded by fossil charcoal and chemical elements, Paleogeogr. Paleoclimatol. Paleoecol., 305(1–4), 329–336, https://doi.org/10.1016/j.palaeo.2011.03.015, 2011.

Shen, H., Zhou, X., Zhao, K., Betts, A., Jia, P.W., and Li, X.: Wood types and human impact between 4300 and 2400 yr BP in the Hexi Corridor, NW China, inferred from charcoal records, Holocene, 28(4), 629–639, https://doi.org/10.1177/0959683617735586, 2018.

The pollen records in Fig 4 are from the archaeological sites? The captions of this figure should be more explained. If so, the similar changes in the records of charcoal and pollen do not indicate the woody degradation in northwestern China. Because pollen records from the section of archaeological site likely also show the trend of wood usage.

Yes, the pollen diagrams in Fig 4 are from archaeological sites, while the strata contain both natural layers before humans became prominent in the area and cultural layers under human influence, as showed in Fig 4a (see the column chart on the left side). Therefore, the pollen records could provide clues to not only the initial natural vegetation but also the wood usage and vegetation variation during human settlement. In this study, the pollen records from Donghuishan, Majiayao, and Xishanping all show a relatively med-level abundance of *Picea* in natural layers before humans. With the beginning of human occupations, both wood charcoal and pollen content were relatively high, showing intensive collection of spruce wood, as the reviewer suggested. While afterwards, wood charcoal and pollen both decreased quickly, and the content of pollen is less than that in the natural layer, indicating the decline of spruce forest. At the later stages, people kept using spruce wood, while the pollen percentage remained low and almost disappeared in Donghuishan and Majiayao sites, suggesting that spruce forests around the sites were largely destroyed. Thus, we infer that the synchronous decline of wood charcoal and pollen in these sites suggested not only the pattern of wood usage, but also a quick deforestation process of spruce forest due to intensive human cutting.

We will also add more information to the figure caption, as the reviewer suggests.

---

## Author Comment (AC2)

This MS reviewed the published data of wood utilization in both Gansu and Xinjiang Provinces, and also incoporated their own works, to show the human-environmental interaction, and adaption in the regions of the Pre-Silk road. This paper is informative, well written and well illustrated. There are also some sentences should be reconsidered and updated to make this paper into a better version.

Many thanks for your time, and your suggestions are much valuable for improving our manuscript.

Line 94, both barley and wheat, their Latin names should be added.

The Latin names have been added.

Line 251. Usually, it is difficult to identify the wood into species level. Since there are several wood sepcies in NW China, It the wood of *Elaegnus angustifolia* different from the relative species in the same genus?

We agree with the reviewer that it is difficult to identify wood charcoal into species level. According to Flora of Xinjiang (Editorial Committee of Flora of Xinjiang, 2011), two species, namely *Elaeagnus angustifolia* and *Elaeagnus oxycarpa*, are distributed in Xinjiang. Now, *Elaeagnus oxycarpa* is classified into *Elaeagnus angustifolia* (https://powo.science.kew.org/taxon/urn:lsid:ipni.org:names:70028324-1), suggesting that only the species of *Elaeagnus angustifolia* is native to Xinjiang. Thus, we think the wood charcoal from our study sites in Xinjiang should be *Elaeagnus angustifolia.* In order to make our readers unconfused about this part, we will change the identification result into *Elaeagnus.*
Reference:
Editorial Committee of Flora of Xinjiang. (Eds.): Flora of Xinjiang (Volume 3), Xinjiang Science and Technology Press, Urumqi, ISBN 9787546608341, 2011.

Line 319 to 320. A wide conception of *Prunus*, that is, *Prunus sensu lato*(*s.l.*)includes

*Prunus*, *Amydalus*, *Cerasus*,*Padus*, as well as *Armeniaca*.

Changes are made as suggested.

Line 390-391. The Latin name of the five cereal species should be added behind.

The Latin names have been added.

Line 399. The word "Found" could be changed into "discovered".

Change is made as suggested.

Line 424. *P. persic*a should be changed into "*Prunus persica*".

Thanks for this suggestion. Change is made as suggested.

Line 447-450. The references should be supplied for each sentence.

References have been supplied.